# The complete mitochondrial genome of *Cycas debaoensis* revealed unexpected static evolution in gymnosperm species

Sadaf Habib[1,2], Shanshan Dong[2], Yang Liu[2], Wenbo Liao[1]*, Shouzhou Zhang[2]*

**1** School of Life Sciences, Sun Yat-sen University, Guangzhou, China, **2** Fairy Lake Botanical Garden, Shenzhen & Chinese Academy of Sciences, Shenzhen, China

* lsslwb@mail.sysu.edu.cn (WL); shouzhouz@szbg.ac.cn (SZ)

## Abstract

Mitochondrial genomes of vascular plants are well known for their liability in architecture evolution. However, the evolutionary features of mitogenomes at intra-generic level are seldom studied in vascular plants, especially among gymnosperms. Here we present the complete mitogenome of *Cycas debaoensis*, an endemic cycad species to the Guangxi region in southern China. In addition to assemblage of draft mitochondrial genome, we test the conservation of gene content and mitogenomic stability by comparing it to the previously published mitogenome of *Cycas taitungensis*. Furthermore, we explored the factors such as structural rearrangements and nuclear surveillance of double-strand break repair (DSBR) proteins in *Cycas* in comparison to other vascular plant groups. The *C. debaoensis* mitogenome is 413,715 bp in size and encodes 69 unique genes, including 40 protein coding genes, 26 tRNAs, and 3 rRNA genes, similar to that of *C. taitungensis*. *Cycas* mitogenomes maintained the ancestral intron content of seed plants (26 introns), which is reduced in other lineages of gymnosperms, such as *Ginkgo biloba*, *Taxus cuspidata* and *Welwitschia mirabilis* due to selective pressure or retroprocessing events. *C. debaoensis* mitogenome holds 1,569 repeated sequences (> 50 bp), which partially account for fairly large intron size (1200 bp in average) of *Cycas* mitogenome. The comparison of RNA-editing sites revealed 267 shared non-silent editing site among predicted vs. empirically observed editing events. Another 33 silent editing sites from empirical data increase the total number of editing sites in *Cycas debaoensis* mitochondrial protein coding genes to 300. Our study revealed unexpected conserved evolution between the two *Cycas* species. Furthermore, we found strict collinearity of the gene order along with the identical set of genomic content in *Cycas* mt genomes. The stability of *Cycas* mt genomes is surprising despite the existence of large number of repeats. This structural stability may be related to the relative expansion of three DSBR protein families (i.e., *RecA*, *OSB*, and *RecG*) in *Cycas* nuclear genome, which inhibit the homologous recombinations, by monitoring the accuracy of mitochondrial chromosome repair.

**Data Availability Statement:** All relevant data are within the paper and its Supporting information files.

**Funding:** This project is funded by the Biodiversity Survey and Assessment Project of the Ministry of Ecology and Environment, China (No.2019HJ2096001006 to SZ). The funders had no role in the designing the research, data collection, analysis, or manuscript preparation.

**Competing interests:** The authors have declared that no competing interests exist.

# Introduction

Mitochondrial (mt) genomes provide a substantial genetic information for phylogenetic reconstructions and exploration of essential cellular processes. Recent advances in high-throughput sequencing technologies has significantly facilitated the assemblage of plant mt genomes, and analysis of their structural diversity and evolutionary trends [1–3]. Among major land plant groups, mitochondrial genomes of the earliest land plant groups are relatively conserved due to narrow size variation and similar gene content [4, 5]. Conversely, mitogenomes of vascular plants exhibit highly dynamic characters: from 66 Kb in *Viscum scurruloideum* [2] to 11 Mb in *Larix sibirica* [6], with known genes ranging from 19 to 64 excluding duplicate genes and ORFs (Open reading frames), and intron content ranging from 5 in *Viscum* [2] to 26 in ferns and early diverging gymnosperms [7, 8]. Additionally, plant mitogenomes vary significantly in their nucleotide substitution rates, RNA editing site abundance, and the occurrence of repeat-mediated recombinations [1, 2, 9, 10]. Moreover, plant mitochondria also exhibit extensive inter- or intraspecific variation in genome size and structure, resulting from large sequence duplications and frequent rearrangements in angiosperms [11–13]. However, this phenomenon is less explored in gymnosperms.

Research of mitogenome for vascular plants has been focused mostly on angiosperms. In contrast to the well-studied angiosperms, only 11 mt genomes has been reported for gymnosperms to date (as of April. 2021). Gymnosperms, with approximately 1,000 species, are considered as economically and ecologically significant plants as they account for roughly 40% of the world's forests flora [14, 15]. Among the main lineages gymnosperms, mitogenome has representative for each of the five major groups, i.e., cycads: *Cycas taitungensis* [8], ginkgo: *Ginkgo biloba* [9], gnetophytes: *Welwitschia mirabilis* [9], Pinaceae (*Pinus taeda*, [direct submission at NCBI, MF991879.1] *Picea abies* [16], *Picea glauca* [17], *Picea sitchensis* [18], *Pinus lambertiana* [19], *Pinus sylvestris* [https://www.ncbi.nlm.nih.gov/assembly/GCA_900143225.1/], *Larix sibirica* [6]) and Conifers II: *Taxus cuspidata* [20].

Gymnosperm mitogenomes are featured by structural dynamics with 40 [9] to 69 [8] genes, 10 to 26 introns [21] and highly variable intergenic spacer regions [20]. Despite having significant variation among genes and non-coding regions (repeated sequences, introns, and plastid and nucleus derived sequences) in gymnosperms, the range of draft mitogenome size divergence of 346 Kb in *G. biloba* to 11 Mb in *L. sibirica* is primarily due to the unidentified DNA [6] and mechanism of mitogenome expansion is differed among gymnosperms [20]. The phenomenon of inter- or intra-specific variation is less studied among gymnosperms. Mitogenomes of *P. abies*, *P. glauca*, and *L. sibirica* are revealed to be extensively rearranged but with exact gene order is unknown due to their highly fragmented mt-genomes [6, 22]. Among gymnosperms, study of interspecific variations in earliest diverging group would improve the entire view on the evolutionary pattern of its mechanism in gymnosperms.

Structural rearrangements among mt genomes are usually related to the abundance of repeated sequences as they can lead to the translocations and inversions of varying stoichiometry by mediating intragenomic homologous recombinations [2, 3, 23]. Among the three earliest nonvascular land plant lineages, absence of repeated sequences in moss mitogenomes and the lack of rearrangements [24], or the presence of repeats and rearrangements in hornworts [25] support this hypothesis. However, liverworts showed somewhat inconsistent pattern in having repeats but with low frequency of recombination [4]. On the other hand, mitogenomes of vascular plants are rich in repeated sequences, which explain their structural lability, with many rearrangements observed even among the inter-familial, inter or infra-generic species [24]. However, inter-specific gene order rearrangements have never been tested in gymnosperms due to the fewer availability of gymnosperm mt genomes and needs further

explorations for a comprehensive understanding of the evolution and diversification of gymnosperm mitogenomes.

Cycads (Cycadales) along with *G. biloba* (Ginkgoales) form the earliest diverging clade of gymnosperms, and is sister to all other gymnosperms [26]. *G. biloba* is a sole species of Ginkgoales, hence cycads are an appropriate group for investigating the ancestral condition and structural stability of mitochondrial genome in gymnosperms. Moreover, *Cycas* mitogenome was among the richest in repeated sequence in mitogenome [20, 24], therefore, it is a perfect candidate to demonstrate the amplitude of genome rearrangement in gymnosperms, and among closely related species in different lineages of land plants, where the genome embarked on a path of radical structural evolution, among all eukaryotes. Cycads are contemporary relic gymnosperm that has been originated before the mid-Permian, and were in their splendor during the Jurassic–Cretaceous [27, 28]. Currently, relics of these enigmatic plants are distributed in the tropical and subtropical regions of the world [28, 29].

Here, we present the complete mitogenome of *C. debaoensis*, a cycad species endemic to the Guangxi region in southern China, to test stability of gymnosperm mitogenome at inter-species level by comparing it with the available mitogenome of *C. taitungensis*. Moreover, our study will elucidate the factors affecting structural rearrangements and genetic basis underlying, i.e., the nuclear surveillance of double-strand break repair (DSBR) protein.

## Material and methods

### Mitochondrial DNA and RNA isolation and mitogenome assembly

The plant tissue of a cultivated *C. debaoensis* tree was collected from Shenzhen Fairy Lake Botanical Garden, Shenzhen, China. No specific permission was required for collection of plant sample used in current study. The sample was identified by Zhang Shouzhou, and the voucher specimen ((No. ZhangSZ2020001) was deposited in SZG (Herbarium of Shenzhen Fairy Lake Botanical Garden, Shenzhen, China). Genomic DNA and RNA was isolated using the CTAB method with modifications described by [30]. The quality and quantity of DNA and RNA were examined using 1% Agarose gel electrophoresis and Qubit fluorometer, respectively. After extraction, 20 μg high-quality DNA were subjected to Nanopore sequencing on an ONT PromethlON 48 platform at Nextonomics (Wuhan, China). About 1 μg of high quality DNA and RNA were fragmented and used to construct paired-end NGS sequencing libraries of insert size 350 and 200 bp, respectively, according to the manufacturer's instructions (Illumina, CA, USA), and then sequenced on an Illumina HiSeq 2000 at NextonOmics Biosciences (Wuhan, China).

The raw genomic and transcriptomic reads were trimmed and filtered for adaptors, low quality and duplicate reads using Trimmomatic (https://github.com/timflutre/trimmomatic). The long Nanopore reads were then *de novo* assembled using NextDenovo (https://github.com/Nextomics/NextDenovo). The raw genome assembly were polished using Illumina paired-end reads using Pilon [31] for three times. The corrected genome assembly was then searched by blast using the previously published mt genome of *C. taitungensis* (AP009381). One mt contig of 527,762 bp was found as a result. Sequencing depth and read coverage of this contig was checked with Illumina DNA-seq reads. The resultant mt contig was overlapped with at the two ends, yielding a circular chromosome of 413,715 bp.

The draft mitogenome of *C. debaoensis* was annotated as previously described by [8, 9]. Briefly, protein coding genes (PCGs) and rRNA genes were annotated by Blastn searches of the non-redundant database at National Center for Biotechnology Information (NCBI) website. The exact gene and exon/intron boundaries were manually adjusted in Geneious v10.0.2, (Biomatters, www.geneious.com) and further corroborated by aligning each gene to its

orthologs from currently available annotated plant mitochondrial genomes at NCBI (www.ncbi.nlm.nih.gov/genome/organelle). The tRNA genes were identified using tRNAscan-SE 2.0 [32]. The annotated *C. debaoensis* mitochondrial genome assembly is deposited to CNGB Sequence Archive (CNSA) of China National GeneBank DataBase (CNGBdb) with accession number CNA0019277, and read mapping file in fastq format was submitted to GenBank under the accession number of SRR13558328. The mitogenome map (Fig 1) was drawn using OGDRAW v1.2. [33].

## Repeats, tandem repeats, Bpu-like elements and plastid derive repeats

Repeats identification of *C. debaoensis* mt genome ($\geq$ 50 bp) was done using ROUSFinder.py script following the procedure of Wynn & Christensen [34], and tandem repeats (TRs) were detected using Tandem Repeats Finder using default parameters [35]. Bpu-like elements and plastid homologous sequences were identified using Blastn (E-value $\leq$ 1e-6, word size = 7), as described by [9]. Briefly, C. *taitungensis* 36 Kb Bpu-like consensus sequence (AAGGTTATCC CTTTCCTGAGCGTAGCGAAGGGAAGG) described by [8] was used as a query to search the Bpu-like elements in *C. debaoensis* and *C. taitungensis*, with 7 mismatches (including gaps) to the 36 Kb Bpu-like consensus sequence (known as dominant type Bpu-like sequence hereafter) were allowed for calculations. For searching of plastid derived (cp) sequences, *C. debaoensis* mt genome was blasted against the available cp genomes of *Cycas* at NCBI. Genes with simultaneous occurrences in cp and mt genome (*atp1/atpA*, *rpl16*, *rps4*, *rrn26/rrn23*, and *rrn18/rrn16*) were not considered.

## Identification of genome rearrangements and DSBR protein analyses

Forty-six representative species of land plants including six gymnosperm mt genomes (excluding the highly fragmented mt-genomes of Pinacea) were selected to determine the inter-generic and inter-species genome rearrangements among gymnosperms and other major land plant groups (S1 Table). The data matrix was constructed based on the order of all PCGs, rRNA and tRNA in the genome along with their transcriptional direction, excluding foreign or non-functional pseudo genes (S1 Table). Duplicated genes and parts of trans-spliced genes were treated independently. The final data matrix was then used to identify genome rearrangements. Pairwise comparison of the mt genomes was conducted using double cut and join (DCJ) model under the likelihood criterion [36], implemented with UniMoG [37]. This program estimates the minimal rearrangement events between a pair of mt genomes. The phylogenetic tree topologies for these representative taxa were drawn from 1KP project [38], and visualized with heatmap of mt genome rearrangements using the online platform OmicShare tools (http://www.omicshare.com/tools/Home/Soft/heatmap).

Genome stability and prohibition of the recombination between repeated DNA sequences was linked to the nuclear encoded double-strand break repair (DSBR) proteins [39]. Relative expansion of these gene family members in mitochondrial genomes can explain the enhance of nuclear surveillance of DSBR protein maintaining the mt genomic structures and function. Characterization of six frequently reported DSBR proteins i.e., *MSH* [40], *RecA* [41, 42]; *RecX* [4], *RecG* [43], *OSB* [44], and the *Whirlies* [45, 46], was conducted following the steps described by [4]. Sixteen vascular plant species with majority of them from gymnosperms were selected for analyses (S2 Table). HMMER [47] was used to perform Hidden Markov Model (HMM) searches at E-value of 1e-6 and alignment length $\geq$ 50%, using Pfam domains of RecA (PF00154; *RecA* gene), RecX (PF02631; *RecX* gene), SSB (PF00436; *OSB* gene), MutS_V (PF00488; *MSH* gene), Whirly (PF08536; *Why* gene), and DEAD and Helicase_C (PF00270, PF00271; *RecG* gene) as query to search the annotated proteins in selected vascular

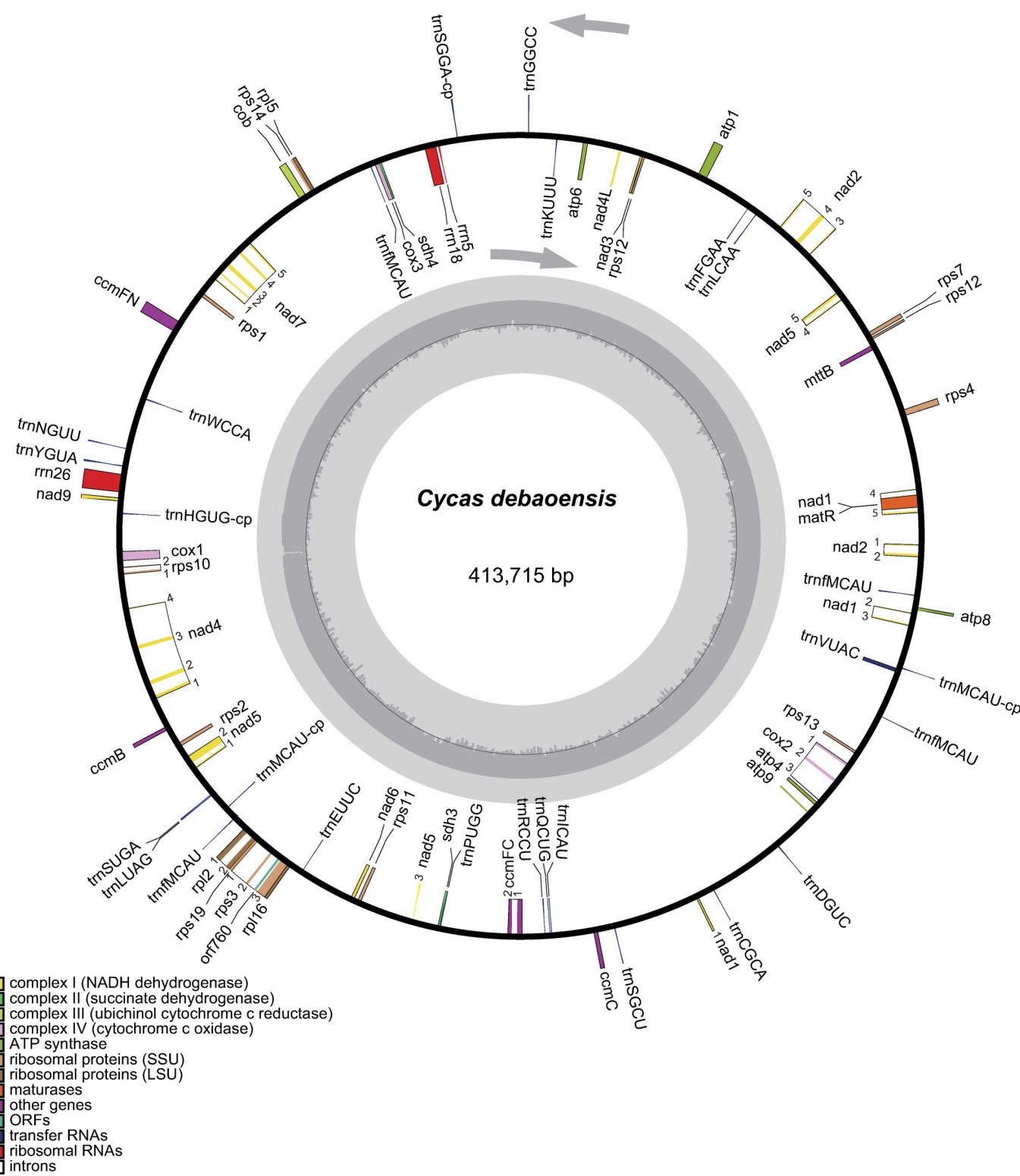

**Fig 1. The map of mitochondrial genome of *Cycas debaoensis*.** Genes (along with exon numbers) shown inside and outside of the circle are transcribed in clockwise and counter-clockwise directions, respectively. Plastid derive tRNAs are indicated with a '-cp' suffix.

plant species. The resultant protein sequences from HMMER search were then validated using the SMART [48] and Pfam [49] databases, and aligned using MAFFT under default parameters [50]. Using maximum likelihood criterion, the final alignment was used to construct the phylogeny in IQ-TREE [51] with 1,000 bootstrap replicates. The subcellular locations of DSBR proteins were predicted using TargetP 1.1 webserver [52] and their homologs were identified at the online database UniProt (https://www.uniprot.org/peptidesearch/).

### RNA editing site identification

RNA editing sites were predicted for *C. debaoensis* protein coding genes (CDS) using the online tool PREP-Mt [53], with the default cutoff score set to 0.2. Availability of high depth RNA-seq data made it possible to calculate the empirical RNA-editing sites on protein coding genes (CDS) in *C. debaoensis*. As PREP-Mt predicts the nonsilent RNA editing sites, we only compare the predicted RNA-editing sites to the empirically annotated RNA-editing sites on CDS of *C. debaoensis*, following the methods described in [54] and [55]. Briefly, RNA-seq clean reads were mapped to the reference genome file containing the CDS of *C. debaoensis* using Tophat2 [56]. The accepted mapping hits in bam format were sorted using Samtools [57] and Bcftools [58]. The resultant vcf file was used to generate the snp file using a perl script (Dryad Digital Repository, accession 10.5061/dryad.nzs7h44ms). Potential genomic SNPs were then removed manually by filtering the RNA editing annotation file against the SNP sites by their positions on the genome sequence. Finally, the annotation file of RNA editing sites was manually checked against transcriptome mapping bam file in Geneious v10.0.2 to acquire exact number of RNA editing sites. The WGS read mapping file and transcriptome mapping bam file have been deposited in the Short Read Achieve (SRA) database of NCBI under the accession number of SRR13558328 and SRR13528745, respectively.

Furthermore, we investigated whether the editing frequencies in gymnosperm species are shaped by selection constraint on genes as suggested by Jobson and Qiu [59]. Three gymnosperm taxa included in the analysis are *C. debaoensis*, *G. biloba* and *T. cuspidata*. We calculate the gene-specific rates of evolution, at both synonymous ($d_S$) and nonsynonymous ($d_N$) sites estimated using CODONML implement in PAML [60]. Editing frequency (%) for each gene is calculated as{(A [number of edited sites]/B [gene length]) × 100}.

## Results and discussion

### Genome size and gene content of *C. debaoensis*

The *C. debaoensis* mitogenome is assembled into a single circular molecule of 413,715 bp (CNGBdb accession: CNA0019277), a size in similar range to *C. taitungensis*, *G. biloba* and *T. cuspidata* with genome size of 414 Kb, 346 Kb and 414 Kb, respectively. However, *W. mirabilis* (978 Kb) and species of Pinaceae (*P. taeda*, 1.19 Mb [https://www.ncbi.nlm.nih.gov/nuccore/NC_039746.1/] *P. glauca*, 5.9 Mb [17]; *P. abies*, 4.3 Mb [16]; *P. sitchensis*, 5.5 Mb [18]; *P. lambertiana*, 3.9 Mb [19]; *P. sylvestris*, 986 Kb [https://www.ncbi.nlm.nih.gov/assembly/GCA_900143225.1/]; *L. sibirica*, 11.7 Mbp [6]) have extremely expanded mitochondrial genomes. We compare the general features of mitogenomes of representative taxa of all five major lineages of gymnosperms (Table 1). *C. debaoensis* mitogenome has a GC content of 46.9% (Table 1), similar to *C. taitungensis* that with a GC content of 46.9%, and lies within a range (i.e., < 50%) of *P. taeda*, early angiosperms [61], and two lycophytes species (*Huperzia* and *Isoetes*). However, GC content of *G. biloba*, *W. mirabilis*, *T. cuspidata*, and ferns mt genomes is found to be > 50%. Consistent with *C. taitungensis*, mitogenome of *C. debaoensis* encodes 69 unique genes, containing 40 protein coding genes, 26 tRNAs, and a same set of 3 rRNA genes (*rrn5*, *rrn16*, and *rrn26*) as in angiosperms (Table 1). The total gene length of *Cycas* is about

**Table 1. General features of mitochondrial genomes of gymnosperms.**

| Features | C. debaoensis | C. taitungensis | G. biloba | P. taeda | W. mirabilis | T. cuspidate |
|---|---|---|---|---|---|---|
| Accession | CNA0019277 | AP009381 | KM672373 | MF991879.1 | KT313400 | MN593023 |
| Size (Kb) | 413 | 414 | 346 | 1,191 | 978 | 468 |
| GC% | 46.9 | 46.9 | 50.4 | 47 | 53 | 50.39 |
| Genes | 69 | 69 | 66 | 58 | 40 | 46 |
| tRNAs | 26 | 26 | 23 | 12 | 8 | 10 |
| rRNAs | 3 | 3 | 3 | 6 | 3 | 4 |
| PCGs | 40 | 40 | 40 | 40 | 29 | 32 |
| Introns | 26 | 26 | 25 | 26 | 10 | 15 |
| Total gene length (Kb) | 87.6 (21%) | 87.6 (21%) | 80.4 (23%) | 72.8 (6%) | 50.6 (5%) | 59 (12%) |
| Protein exons (Kb) | 32.3 (8.4%) | 32.3 (8.4%) | 34 (9.8%) | 33.8 (2%) | 29.7 (3%) | 31 (6.6%) |
| Cis-spliced introns (Kb) | 51.2 (12.3%) | 51.2 (12.3%) | 39.1 (11%) | 26.5 (2.2%) | 5.9 (0.7%) | 6.9 (1.2%) |
| Intergenic spacer (Kb) | 326 (79%) | 326 (79%) | 266 (77%) | 1118 (94%) | 928 (95%) | 410 (88%) |
| Plastid origin (Kb) | 16 (3.8%) | 17 (4.1%) | 0.3 (0.1%) | 5.6 (0.5%) | 7.9 (0.8%) | 0 (0%) |
| Nuclear origin (Kb) | 3.4 (0.8%) | 3.4 (0.8%) | 1.9 (0.6%) | 5.3 (0.5%) | 2.5 (0.3%) | 3.5 (0.8%) |
| Repeats (Kb) | 51.7 (12.5%) | 54.3 (13%) | 32 (9.3%) | 170 (14.2%) | 50 (5.0%) | 62 (13.2%) |
| TRs (Kb) | 23 (5.7%) | 22 (5.3%) | 3.6 (1.1%) | 71 (6.0%) | 24 (2.5%) | 48 (10.2%) |
| Bpu-like elements* | 486 | 504 | 19 | 0 | 0 | 0 |

* *C. taitungensis* dominant type Bpu-like sequence used as query.

87.7 Kb, which accounts for 21% of the total mt genome length, including about 32 Kb (8.4% of total genome length and 36% of total gene length) of protein coding sequences. The protein exon length among observed gymnosperm species lies within the range of 29.7 Kb in *W. mirabilis* (29 PCGs), to 34 Kb in *G. biloba* (41 PCGs), which accounts for only 3% and 9.8% of the total gene length, respectively. Although, there is significant difference in the number of mitochondrial genes, the variation of noncoding DNA content is the major contributor towards large size mitogenomes of gymnosperms [7, 20]. In addition to gene content, other factors associated with mitogenome expansion include foreign sequences, size and number of repeated sequences [10, 62–65]. However, in case of gymnosperms, an ample amount of unidentified DNA denoted to the mitogenome expansion [9, 22], and the above mentioned factors have negligible contribution towards overall mitogenome length variation (Table 1).

## Evolution of intron content of *Cycas* and across land plants

The evolution of intron content is conservative with comparison of the two *Cycas* mitochondrial genomes. Twenty-six group II introns disrupt 10 protein coding genes (*ccmFC, cox2, nad1, nad2, nad4, nad5, nad7, rpl2, rps3, rps10*) in mt-genomes in both *Cycas* species (Table 1), including 21 cis-spliced (adding up to 51 Kb and 12.3% of total genome length) and 5 trans-spliced introns (i.e., *nad1i394, nad1i669, nad2i542, nad5i1455,* and *nad5i1477*). The early diverging *Pinus* and *Cycas* mt genomes retain ancestral intron content of seed plants (26 introns). However, in addition to five shared trans-spliced introns, eight more introns are shifted from cis- to trans- spliced in *P. taeda, Picea abies* and *Picea glauca* mt-genomes (Fig 2). Intron content of *G. biloba* (30 Kb; 11%) only differs from *Cycas* owing to the loss of one intron (*rps10i235*) in it (Table 1). The intron content is greatly declined in subsequent lineages, e.g., *T. cuspidata* and *W. mirabilis* maintain 15 and 10 introns, respectively (Fig 2). Most of the seed plant mitochondrial introns are first evolved in ferns as *Psilotum* and *Ophioglosum* share 23 and 19 introns with seed plants, respectively. However, trans-splicing of *nad1i394,*

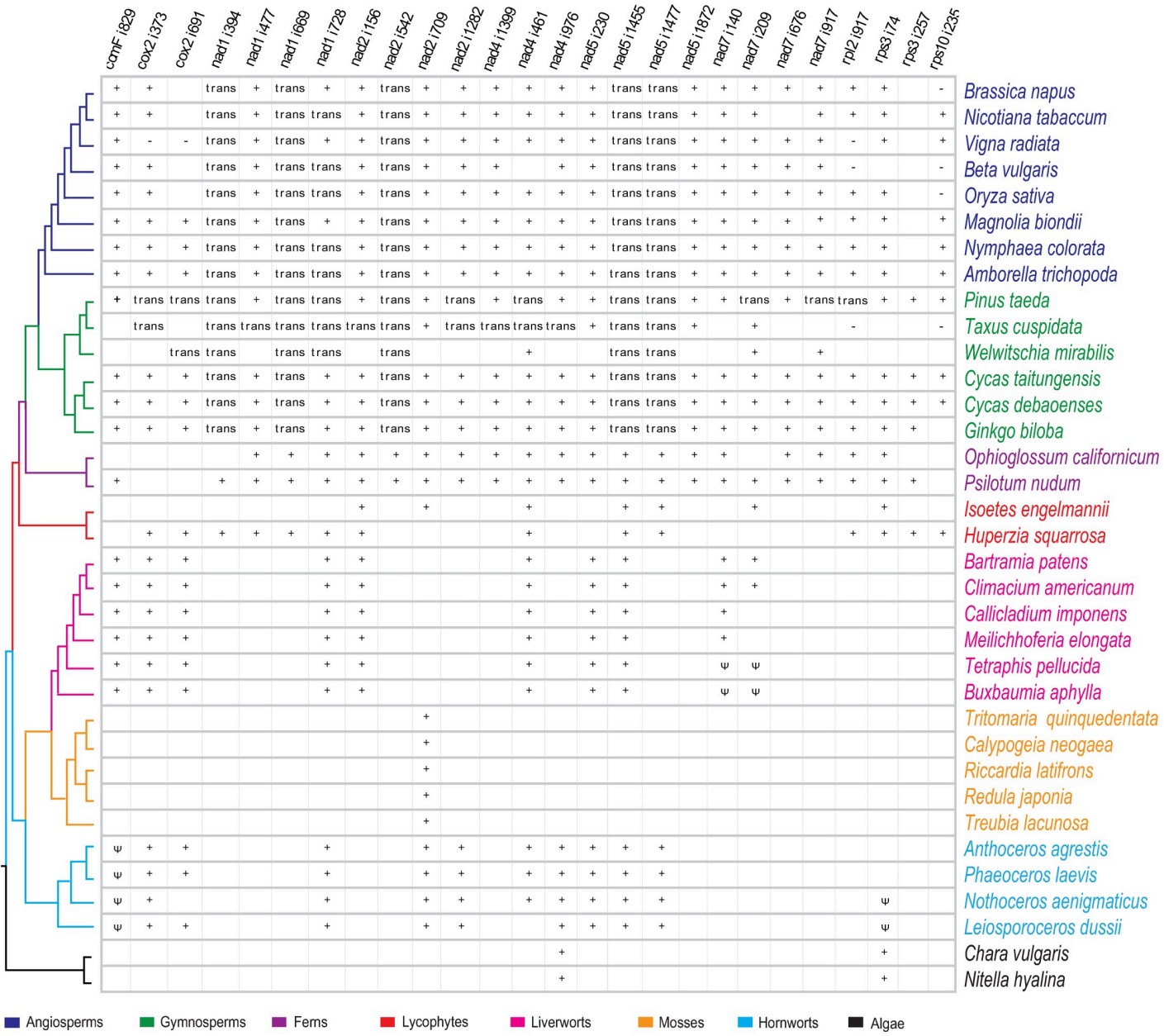

**Fig 2. Distribution pattern of 26 ancestral seed plant introns among land plants.** +, *trans*,—and Ψ indicates presence of an intron, trans-splicing intron, missing gene and pseudogenization, respectively. Phylogenetic tree is inferred from 1KP project [38].

*nad1i669*, *nad2i542*, *nad5i1455*, and *nad5i1477* introns predominantly occurs in gymnosperms and angiosperms (Fig 2). Liverworts lack all of these group II introns, except *nad2i709*. However, hornworts and mosses share 10 group II introns with seed plants (Fig 2). Mitochondrial intron content is highly conserved within major lineages of land plants, though varies greatly among them. Overall, intron loss and trans-splicing is most prevalent in gymnosperms in comparison to other land plants. The gradual intron losses from early diverging lineages to the derived ones in gymnosperms may be related to retro-processing events for the introns removed at 3' ends, or selective pressure to retain the introns at near to 3' and 5' ends [20].

## Repeats, tandem repeats and Bpu-like elements

Mitochondrial genome of *C. debaoensis* appears to have 1,569 repeated sequences of longer than 50 bp, adds up to 51.7 Kb in length, i.e., 12.5% of the total mt genome length (Table 1). *C. taitungensis* contains 1,522 repeated sequences of a total of 54.3 Kb in length, which accounts for about 13% of the total genome. Interestingly, no large repeats (> 1 Kb) are found in *C. debaoensis*, compared to *C. taitungensis* having 2 large repeats. Both species share numerous repeats of intermediate (100–900 bp) and short length (< 100 bp). The repeated sequence proportion of gymnosperms covers about 9–14% of the total mt genome length, except for *W. mirabilis* with only 5% of repetitive sequence in its mt genome (Table 1). The proportion of repetitive sequences varies greatly among major land plant groups. The earliest land plants have few (mosses), to an average of 4.5% (liverworts) of repeated sequences in their mt-genomes [4, 24]. On the other hand, ferns (*Psilotum* about 57% of genome; *Ophioglossum* about 40% of genome) and also angiosperm species such as *Nymphaea* (49%) have the most repeat-rich mitogenomes [7, 22].

Repeat insertion events are considered contributing significantly to intron size expansion [7, 66]. Comparing the length of 26 cis-spliced introns found in *Cycas* with four gymnosperms (*G. biloba*, *P. taeda*, *T. cuspidata*, and *W. mirabilis)* and other vascular plant representatives reveals that ferns (*Ophioglossum*, *Psilotum*), *Cycas*, and *P. taeda* have introns of relatively longer in length along with abundant repeated sequences, as compared to lycophytes and angiosperms (Table 2). Intron lengths of *cox2i691*, *nad2i1282*, *nad4i1399*, *nad4i976* and *nad7i676* are > 1000 bp longer, and of *rpl2i917* was > 500 bp longer in *Cycas* than in any of their gymnosperm counterparts (Table 2). Whereas, *ccmFci829*, *rps10i235* and *rps3i257* are > 1000 bp longer in *P. taeda* than all other observed taxa (Table 2). These elongated introns appear to have abundant repeated sequences, which are responsible for longer introns in *Cycas* and *P. taeda* mt genomes. Moreover, *C. debaoensis* mt genome contain 23 Kb (5.7% of total genome length) of tandem repeat sequences (TRs), which is comparable to *C. taitungensis* having 22 Kb (5.3%) of tandem repeats (Table 1). Despite the considerable disparity among TRs proportion in gymnosperms, their impact on overall genome size expansion is trivial.

*C. taitungensis* mitogenome contained abundant short interspersed repetitive elements known as Bpu-like sequences/elements [8]. These mobile elements are characterized by having two conserved terminal direct repeats (AAGG) and a recognition site for the restriction endonuclease, known as Bpu10I (CCTGAAGC; nt 15–21). We retrieve 486 variants of Bpu-like elements in *C. debaoensis* using the dominant type Bpu-like sequence as a query. Among these variants, 251 Bpu-like sequences show 100% identity to the dominant 36 bp Bpu-like sequence. Another 41 sequences are 100% identical to the dominant Bpu-like sequence, but with reduced sequence length of 29–35 bp (S3 Table). Using the same parameters, *C. taitungensis* is found to have 504 variants of Bpu-like elements, 309 of them are 100% identical to the dominant type 36 bp Bpu-like sequence (S3 Table). Moreover, another 36 Bpu-like elements of 30–35 bp are 100% identical to the dominant type Bpu-like sequence. Bpu-like insertion sites for *C. debaoensis* and *C. taitungensis* are mostly found to be orthologous (S3 Table). Using the same parameters, we blast the Bpu-like-elements against the *G. biloba*, *T. cuspidata*, *W. mirabilis* and *P. taeda* mt genomes. *G. biloba* is found to have 19 Bpu-like sequence with only one (35 bp) of them showing 100% identity to the dominant type of Bpu-like elements in *Cycas* (Table 1). All the other *Ginkgo* Bpu-like variants are differed from the dominant *Cycas* Bpu-like sequence at position 9 (C to A), 17 (T to C) and 28 (A to G), but have conserved terminal repeats (position 1–4 and 33–36) and Bpu10I endonuclease recognition site (position 15–21) similar to *Cycas*. Using this *Ginkgo* Bpu-like consensus sequence (AAGGTTAT**A**CCTTTCC **C**GAGCGTAGCG**G**AGGGAAGG), nearly 100 variants of Bpu-like elements are identified in

**Table 2. Intron size variations among vascular plant mitogenomes.**

| | Intron | Lycophytes | | Ferns | | Gymnosperms | | | | | Angiosperms | |
|---|---|---|---|---|---|---|---|---|---|---|---|---|
| | | *Huperzia* | *Selaginella* | *Ophioglossum* | *Psilotum* | *Cycas* | *Ginkgo* | *Pinus* | *Taxus* | *Welwitschia* | *Liriodendron* | *Phoenix* |
| 1 | ccmFci829 | – | – | – | 4328 | 1063 | 1003 | 2609* | – | – | 1042 | 949 |
| 2 | cox2i373 | 2318 | Ɵ | – | – | 2973 | 2662 | Ɵ | Ɵ | – | 2752 | 1289 |
| 3 | cox2i691 | 2528 | 2111 | – | – | 3991* | 2722 | Ɵ | – | Ɵ | 2220 | 1596 |
| 4 | nad1i394 | 1673 | 3130 | – | 5451 | Ɵ | Ɵ | Ɵ | Ɵ | Ɵ | Ɵ | Ɵ |
| 5 | nad1i477 | 1490 | 5503 | 2998 | 4736 | 1873 | 1501 | 2,157 | Ɵ | – | 1419 | 1431 |
| 6 | nad1i669 | 3108 | 2963 | 4590 | 4325 | Ɵ | Ɵ | Ɵ | Ɵ | Ɵ | Ɵ | Ɵ |
| 7 | nad1i728 | 3402 | 4061 | 3287 | 3860 | 3945 | 3672 | Ɵ | Ɵ | Ɵ | 4748 | Ɵ |
| 8 | nad2i1282 | – | – | 2159 | 3196 | 3491* | 2355 | Ɵ | Ɵ | – | 1453 | 1387 |
| 9 | nad2i156 | 1518 | 5175 | 2174 | 4537 | 2093 | 1450 | 1536 | Ɵ | – | 1444 | 1234 |
| 10 | nad2i542 | – | 1322 | 1913 | 4978 | Ɵ | Ɵ | Ɵ | Ɵ | Ɵ | Ɵ | Ɵ |
| 11 | nad2i709 | – | 1546 | 1803 | 1510 | 2433 | 2390 | 2015 | 2,364 | – | 2618 | 2487 |
| 12 | nad4i1399 | – | 2779 | 2704 | 7382 | 5452* | 3438 | 1820 | Ɵ | – | 2899 | 2807 |
| 13 | nad4i461 | 2921 | 2004 | 5060 | 2120 | 2080 | 1810 | Ɵ | Ɵ | 1350 | 1404 | 1370 |
| 14 | nad4i976 | – | 4298 | 3025 | 6688 | 5707* | 4382 | 3510 | Ɵ | – | 4413 | 3767 |
| 15 | nad5i1455 | 2908 | 2486 | 3271 | 3879 | Ɵ | Ɵ | Ɵ | Ɵ | Ɵ | Ɵ | Ɵ |
| 16 | nad5i1477 | 2870 | 2314 | 1432 | 4121 | Ɵ | Ɵ | Ɵ | Ɵ | Ɵ | Ɵ | Ɵ |
| 17 | nad5i1872 | – | – | 623 | 549 | 912 | 1233 | 844 | 804 | – | 1316 | 945 |
| 18 | nad5i230 | – | – | 4366 | 7669 | 899 | 870 | 1,177 | 1,293 | – | 845 | 861 |
| 19 | nad7i140 | – | 642 | 1688 | 2904 | 1183 | 1017 | 1558 | – | – | 875 | 856 |
| 20 | nad7i209 | – | 1809 | – | 3495 | 1585 | 1633 | Ɵ | 2,461 | 2144 | 1754 | 1766 |
| 21 | nad7i676 | – | 609 | 1092 | 3487 | 2420* | 1137 | 1105 | – | – | 1013 | 999 |
| 22 | nad7i917 | – | 1656 | 5564 | 8206 | 1938 | 1861 | Ɵ | – | 2431 | 1904 | 1802 |
| 23 | rpl2i917 | 747 | – | 1769 | 777 | 1344* | 782 | Ɵ | # | – | 1471 | 1622 |
| 24 | rps10i235 | 819 | – | – | – | 883 | – | 2,064* | # | – | 838 | 1137 |
| 25 | rps3i257 | 1108 | – | – | 1667 | 1899 | 984 | 2,985* | – | – | – | – |
| 26 | rps3i74 | 3030 | – | 6506 | 1935 | 2980 | 2281 | 3199 | – | – | 1878 | 1832 |

–, # and Ɵ indicate the missing intron, missing gene and trans-spliced introns respectively. Whereas

* indicates the significant intron expansion with evidence of repeat abundance in *Cycas debaoensis* and *Pinus taeda* mt genomes.

*G. biloba* [9]. Bpu-like elements are missing in other gymnosperm mitogenomes (Table 1). These results confirm the expansion of Bpu-like elements only in *Cycas* and *G. biloba*. Based on the most recent phylogenetic reconstruction of cycad [67], *C. taitungensis* belonged to the earliest diverging Clade I (Sections Panzhihuaenses and Asiorientales), and *C. debaoensis* was classified to be part of partially supported clade II (The core Stangerioides clade). Comparing the two *Cycas* mt genomes, Bpu-like elements appears to be static in *Cycas* evolution, possibly since the divergence between cycads and ginkgo, which might imply some functional significance.

## Genome structural evolution across land plants, and repeat-triggered recombinations in *Cycas*

We explore the inter-species and inter-generic mitogenome rearrangement across the major land plant groups along with the number of repeats, and then plotted the results beside a phylogenetic gradient of land plants. It is generally believed that repeated sequences (≥ 50 bp) within the mitochondrial genome create opportunities for intragenomic recombination [23, 40, 68]. Such events involve a crossing-over via homologous recombination between repeated

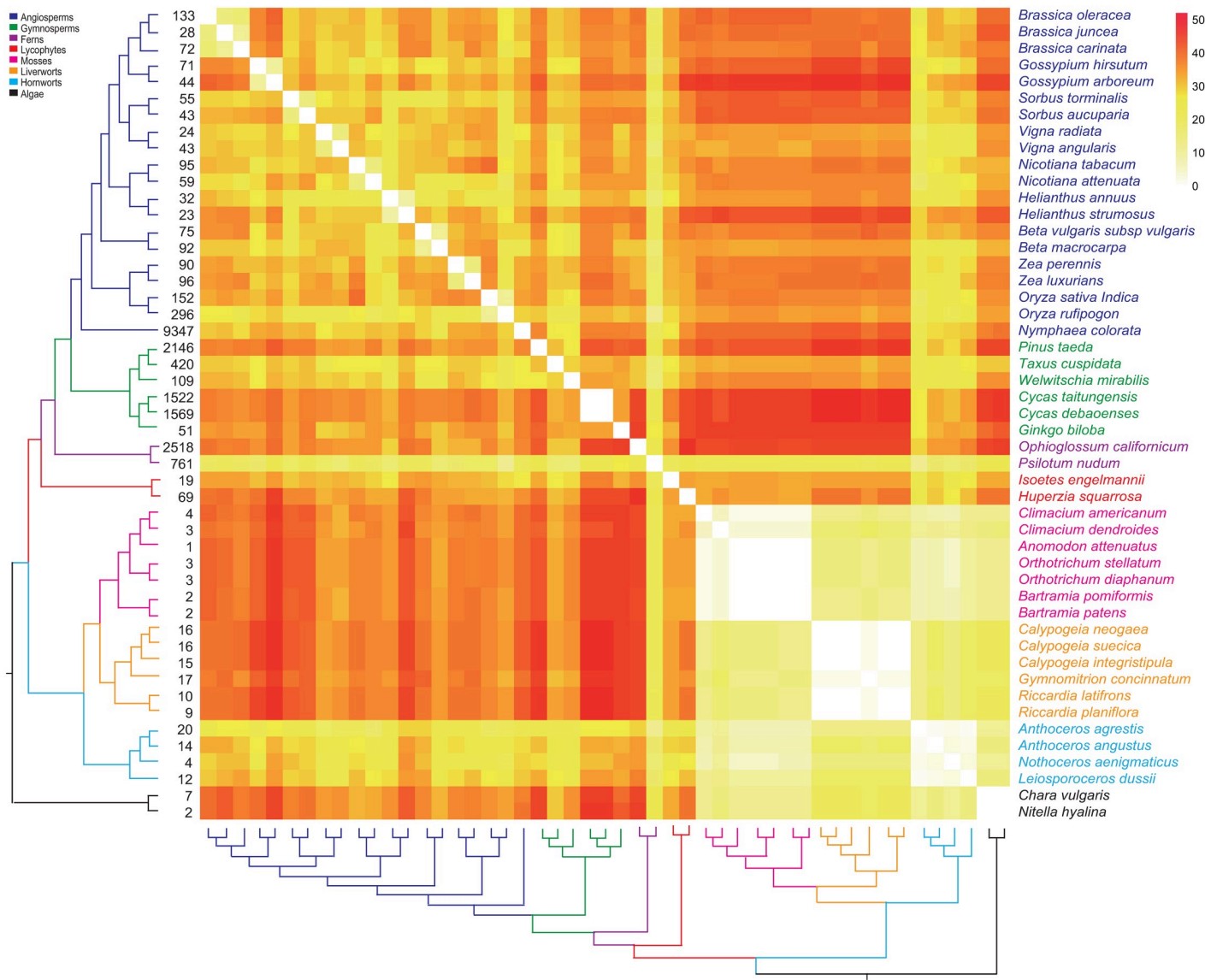

**Fig 3. Heat map of mitochondrial gene order rearrangements in pairwise comparison of 46 representative taxa of major land plant groups along with phylogenetic tree based on the 1KP project [38].** The number of repeats detected for each species are listed beside the tree.

sequences inside a circular genome [39], and result in a novel genome structure. Inter-species and inter-generic rearrangements within bryophytes are not much prominent due to their relatively conserved mitogenomes with fewer number of repeats. Mosses with fewer repeated sequences required 2–4 and 6 inter- and infra-generic rearrangements, respectively. Liverworts with repeated sequences of intermediate abundance among bryophytes, have nearly static mitogenomes. Hornworts appeared to have more number of repeats than mosses and liverworts. Only one inter-specific translocation event occurred among two *Anthoceros* species, and a maximum of 3 inter-generic rearrangements among the observed taxa (Fig 3; S4 Table). Hence, the relationship between number of repeats and genomic rearrangements is not fully supported in bryophytes. However, in angiosperms, inter-specific rearrangements are reported in all of the observed taxa with fairly large number of repeats (Fig 3). Angiosperms

require 11 rearrangements on average to get collinear gene order at infra-generic level. Surprisingly, despite having large number of repeats (>1500), mitogenomes of *C. debaoensis* and *C. taitungensis* share exactly a same gene order. Whereas, 34, 44, 32, and 34 rearrangements are required by *Cycas* mt genomes to get complete collinearity with that of the *G. biloba*, *P. taeda*, *T. cuspidata*, and *W. mirabilis*, respectively (Fig 3; S4 Table). In pairwise comparison, gymnosperm mitogenome require a minimum of 27 (between *P. taeda* and *W. mirabilis*) to a maximum of 44 (between *P. taeda* and *Cycas*) rearrangements to get the same gene order. Any two gymnosperm mitogenome varies by 31 rearrangements on average (Fig 3; S4 Table). The strict collinearity among the gene order along with the identical set of genomic content in *Cycas* mt genomes confirm their structural stability, lack of recombination, and support the hypothesis that repeats may not be sufficient for recombination to occur within the mt genomes [4].

Genomic rearrangements in mt genomes presumably accompany by cis- to trans-spliced intron transitions. Bryophytes have fewer or no rearrangements in each major group, and lacks the trans-spliced introns in their mitogenomes (Figs 2 and 3). However, *Sleginella moellendorffii* (lycophyte) has extensively rearranged mitochondrial genome and with four transspliced introns [13]. Conversely, angiosperm mitogenomes with numerous trans-spliced introns show extensive DNA rearrangement (Figs 2 and 3). Among gymnosperms, our analyses of genomic rearrangements indicate that *P. taeda* with highest number of trans-spliced intron (13 trans-spliced introns) need 44 and 42 rearrangements to get collinearity with early diverging *Cycas* and *G. biloba*. Furthermore, *Picea* mt genomes are also appeared to be highly recombinogenic [22] along with extensive trans-splicing events. Thus, the occurrence of transsplicing with recombinogenic mitogenomes in gymnosperms suggests that shifts from cis- to trans-splicing in plant mitochondria is mainly caused by genomic rearrangements [21].

## Nuclear surveillance of mt genome stability of *Cycas*

The overall stability in gene content and genome structure in *Cycas* mt genomes is significant. The structural stability of mt genomes is related to the nuclear encoded DSBR genes, which hinders the homologous recombinations, by monitoring the accuracy of mitochondrial genome repair [39]. Six frequently reported DSBR genes include the *RecA*, OSB, MSH, *RecX*, Why, and *RecG* [4, 40, 43–45]. We have screened the exemplars of the major vascular plant groups along with seven representatives from major gymnosperm lineages (S2 Table) for these genes and gene families, and analyze their copy numbers in gymnosperms. Phylogenetic reconstruction of six DSBR genes reveals relatively higher expansion of three DSBR protein families (*RecA*, OSB, and *RecG*) in *C. debaoensis* nuclear genome (S1 Fig). Similar set of DSBR proteins are found to be expanded in liverworts, causing their compact mt genomes [4]. These DSBR proteins also show considerable expansion in *G. biloba* and *P. taeda* as compared to other land plant representatives, which suggests possible existence of mt genome stability within these two groups. *G. biloba* also found to have limited repeat mediated recombinational activity [9], which indicates that these nuclear encoded proteins perform a certain level of recombination surveillance, controlling homologous recombination within the mitogenome. Although subcellular localization and *in vivo* function of some of these DSBR proteins needs further investigation, the notable expansion of these protein families among compact mt genomes such as in liverworts [4] and *Cycas* (present study) cannot be completely neglected.

## Plastid derived sequences

The *C. debaoensis* mitogenome possesses 22 plastid derived insertions ranging from 62 bp to 2,707 bp (Table 3) encompassing the total length of 16 Kb, which accounts for 3.8% of total mt

**Table 3. Plastid insertions in the mitochondrial genome of *Cycas debaoensis*.**

| Insertion | Identity (%) | Length | Start | End | Plastid genes | E-value | Bit score |
|---|---|---|---|---|---|---|---|
| 1 | 92.796 | 2707 | 388082 | 390750 | None | 0 | 3847 |
| 2 | 92.737 | 1308 | 390781 | 392071 | None | 0 | 1855 |
| 3 | 89.939 | 1471 | 110970 | 112426 | *ycf3** | 0 | 1851 |
| 4 | 95.951 | 1062 | 393537 | 394582 | *atpB** | 0 | 1709 |
| 5 | 84.731 | 1670 | 112464 | 114041 | *ycf3*-trnS* | 0 | 1561 |
| 6 | 88.306 | 1334 | 203429 | 204727 | *psbA** | 0 | 1554 |
| 7 | 97.308 | 743 | 395525 | 396259 | *rbcL** | 0 | 1254 |
| 8 | 95.979 | 746 | 392139 | 392872 | *atpE*-atpB** | 0 | 1201 |
| 9 | 98.353 | 607 | 392902 | 393508 | *atpB** | 0 | 1064 |
| 10 | 91.871 | 775 | 387315 | 388082 | *ndhC** | 0 | 1059 |
| 11 | 97.373 | 609 | 394883 | 395491 | *rbcL** | 0 | 1035 |
| 12 | 87.262 | 683 | 205250 | 205928 | *matK** | 0 | 754 |
| 13 | 87.884 | 553 | 114933 | 115459 | None | 6.60E-176 | 617 |
| 14 | 88.024 | 501 | 202625 | 203116 | *trnHGUG* | 3.14E-159 | 562 |
| 15 | 88.475 | 295 | 204916 | 205204 | *trnKUUU** | 1.60E-92 | 340 |
| 16 | 97.674 | 172 | 394651 | 394822 | *rbcL** | 3.51E-79 | 296 |
| 17 | 85.507 | 207 | 114716 | 114915 | None | 6.09E-52 | 206 |
| 18 | 93.636 | 110 | 205988 | 206097 | *matK** | 1.03E-39 | 165 |
| 19 | 86.885 | 122 | 203240 | 203361 | *psbA** | 2.91E-30 | 134 |
| 20 | 94.118 | 85 | 255735 | 255819 | *trnMCAU* | 3.77E-29 | 130 |
| 21 | 94.595 | 74 | 391862 | 391793 | *trnMCAU* | 3.80E-24 | 113 |
| 22 | 91.935 | 62 | 25479 | 25539 | *trnMCAU** | 8.28E-16 | 86.1 |

* represents partial sequence.

genome length. These plastid-derived sequences are similar to *C. taitungensis* plastid insertions with slight variation in length (i.e., 17 Kb; 4%). These plastid insertions include three functional tRNAs *trnHGUG*, *trnMCAU* (2 copies), *trnSGGA*, and nonfunctional fragments of seven protein coding genes. In other gymnosperms, plastid derived sequences have very little (< 1%) to no (in *T. cuspidata*) contribution towards the genome length. Plastid insertion are less common among early diverging land plant groups, such as bryophytes [4], lycophytes [69], and ferns [7]. In contrast, angiosperms typically have higher proportion of plastid derived sequences, their earliest diverging groups contain 13 Kb (*Nymphaea*) to 138 Kb (*Amborella*) of plastid insertions [10, 66]. In monocots, plastid derived sequences range from 22 Kb in *Oryza* [70] to 24 Kb in *Zea* [71]. Eudicots appeared to have relatively fewer plastid DNA sequences such as with 4.4 Kb in *Arabidopsis* [72], 2.1 Kb in *Vigna* [64], and 7.7 Kb in *Beta* [73]. The relative proportion of plastid derived sequences to the whole genome length in early diverging angiosperms and monocots is similar to *Cycas* (3 to 6%), however, eudicots contain low percentage (< 2%) of plastid derived sequences similar to derived lineages of gymnosperms [9, 20, 66]. Overall, this pattern highlighted that the origin of plastid derived sequences in plant mitochondrion most likely to be appeared in ancestors of vascular plants, expand in early diverging lineages and begin to decline laterally in more derived groups.

## RNA editing in *Cycas*

Using *in silico* prediction method, a total of 1,181 non-silent RNA editing sites are discovered in proteins coding genes of *C. debaoensis*. However, RNA-seq reads mapping identify only

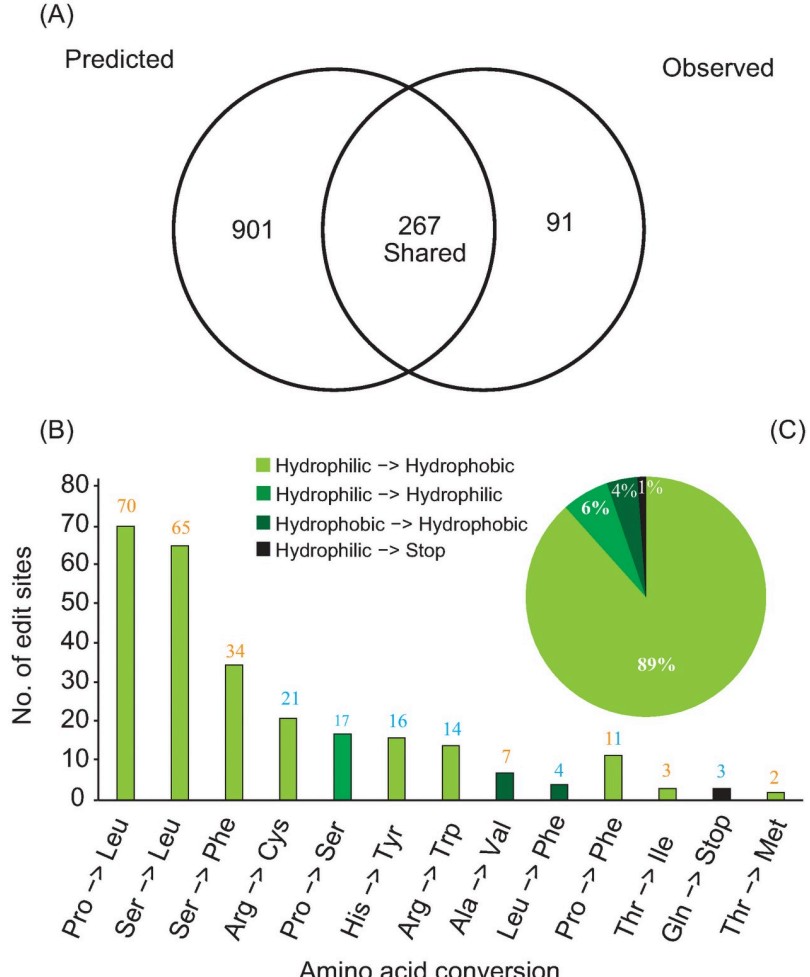

**Fig 4. RNA-editing in *Cycas debaoensis*.** A) Comparison of the number of predicted (PREP-Mt) vs. empirically observed (transcriptome) non-silent RNA editing events. B) No. of RNA editing sites with the amino acid conversion statistics. Number of editing events contributed to amino acid change mention on each bar. Blue and orange color represent the 1st and 2nd codon position responsible for amino acid conversion, respectively. C) Codon alteration proportions according to the hydrophobic and hydrophilic properties of the resulting amino acids.

358 RNA-editing sites, and 267 editing sites are shared between the predicted and empirically annotated editing sites, all of them are C- to -U editing (Fig 4A; S5 Table), indicating high discrepancy between predicted and empirical editing sites. RNA editing sites appear with highest chance at 2nd codon position with 190 editing sites followed by 77 editing sites positioned at 1st codon. As only non-silent editing sites are predicted with PREP-Mt, we manually check 87 unique empirically detected editing sites to identify the number of silent mutations. Thirty-three silent editing sites are recovered, with 31 of them occur at 3rd codon positions (S6 Table). Overall, we confirm 300 editing sites in protein coding genes of *C. debaoensis* with editing site abundance of 10.3%, 63.3% and 26.3% at 1st 2nd and 3rd codon position, respectively. Amino acid changes from non-silent editing events mainly involved Pro → Leu (70), Ser → Leu (65), and Ser → Phe (34), which results in increase of hydrophobicity of these amino acids (Fig 4B). In total, 89% of editing events of amino acid conversion are from hydrophilic → hydrophobic (Fig 4C), which is important for stabilization and functionalization of protein structures [74], and protein-protein interfaces [75]. Furthermore, we

found that membrane-bounded and soluble protein coding genes have experienced similar selective pressures as there is no clear pattern among editing efficiency and gene substitution rates ($d_N$ and $d_S$ vs. editing efficiency %) of three observed gymnosperm taxa (S7 Table). Empirical data regarding abundance of RNA-editing sites in gymnosperm is limited [20]. Future studies with expanded taxon sampling covering major gymnosperm lineages need to be conducted to study the phylogenetic distribution and broad impact of selection based evolution of RNA editing in gymnosperms.

## Conclusion

We assembled the mitochondrial genome of *Cycas debaoensis* and compared it with *Cycas taitungensis*, mt genome of representative gymnosperms, and other major land plant lineages. Our results confirmed that mitogenome of *Cycas* are highly conserved in both gene content and gene order. The stability of *Cycas* mt genomes and lack of recombinations is unexpected in the case of their highly repetitive mt genomes. These repeated sequences significantly contributed to the fairly large size of introns. In addition, we revealed that the stability of *Cycas* mt genome is positively correlated to the expansion of three DSBR protein families in *Cycas* nuclear genome.

## Supporting information

**S1 Fig. Phylogenetic trees of six DSBR protein sequences from 16 vascular plants taxa inferred by Iqtree.** * and # indicate the position of cycad species i.e., *Cycas debaoensis* and *Cycas panzhihuaensis*, respectively.
(PDF)

**S1 Table. Comparison of gene content and gene order from 46 selected land plant mitochondrial genomes.**
(TXT)

**S2 Table. List of 16 vascular plant species for DSBR protein identification.**
(XLSX)

**S3 Table. Bpu-like elements observed in mitogenome of *Cycas debaoensis* and *Cycas taitungensis*.**
(XLSX)

**S4 Table. Gene order rearrangements among representative species of major land plant groups.**
(XLSX)

**S5 Table. RNA-editing sites shared among predicted vs. empirically observed in protein coding regions of *Cycas debaoensis*.**
(XLSX)

**S6 Table. Silent RNA-editing sites empirically observed in protein coding regions of *Cycas debaoensis*.**
(XLSX)

**S7 Table. Gene specific rates of evolution of soluble (Italicized) and memberane-bounded protein coding genes, at both synonymous ($d_S$) and nonsynonymous ($d_N$) sites, with total RNA editing frequency.**
(XLSX)

## Acknowledgments

We are grateful to Yang Peng and Na Li at the Shenzhen Fairylake Botanical Garden for the lab assistances and technical support.

## Author Contributions

**Conceptualization:** Yang Liu, Shouzhou Zhang.

**Data curation:** Sadaf Habib, Shanshan Dong.

**Formal analysis:** Sadaf Habib, Shanshan Dong.

**Funding acquisition:** Shouzhou Zhang.

**Investigation:** Sadaf Habib, Shanshan Dong, Yang Liu, Wenbo Liao.

**Methodology:** Sadaf Habib.

**Project administration:** Shanshan Dong, Yang Liu, Wenbo Liao, Shouzhou Zhang.

**Resources:** Shouzhou Zhang.

**Software:** Sadaf Habib, Shanshan Dong.

**Supervision:** Yang Liu, Wenbo Liao, Shouzhou Zhang.

**Validation:** Shanshan Dong, Yang Liu, Wenbo Liao, Shouzhou Zhang.

**Writing – original draft:** Sadaf Habib.

**Writing – review & editing:** Yang Liu, Wenbo Liao, Shouzhou Zhang.

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
