## [Decision Letter · Decision Letter 0]

9 Jun 2021

PONE-D-21-15488

The complete mitochondrial genome of Cycas debaoensis revealed un expected static evolution in gymnosperm species

PLOS ONE

Dear Dr. Zhang,

Thank you for submitting your manuscript to PLOS ONE. After careful consideration, we feel that it has merit but does not fully meet PLOS ONE’s publication criteria as it currently stands. Therefore, we invite you to submit a revised version of the manuscript that addresses the points raised during the review process.

We look forward to receiving your revised manuscript.

Kind regards,

Tzen-Yuh Chiang

Academic Editor

PLOS ONE

Journal Requirements:

Reviewers' comments:

Reviewer's Responses to Questions

**Comments to the Author**

1. Is the manuscript technically sound, and do the data support the conclusions?

Reviewer #1: Yes

2. Has the statistical analysis been performed appropriately and rigorously? 

Reviewer #1: Yes

3. Have the authors made all data underlying the findings in their manuscript fully available?

Reviewer #1: Yes

4. Is the manuscript presented in an intelligible fashion and written in standard English?

Reviewer #1: Yes

5. Review Comments to the Author

Reviewer #1: Review of “The complete mitochondrial genome of Cycas debaoensis revealed unexpected static evolution in gymnosperm species”

It is nice to have the second Cycas species sequenced for the mitochondrial genome, as it provides data to gain insights into mitogenome evolution in closely related species in this early diverging lineage of gymnosperms. The authors have done a good job of describing the genome structure and content, and have also done a somewhat informative comparison on genome content between cycads and other land plants.

One major point that the authors have failed to appreciate is that the second Cycas mitogenome provides an excellent opportunity to demonstrate the amplitude of genome rearrangement among closely related species in different lineages of land plants, where the genome embarked on a path of radical structural evolution, among all eukaryotes. Until now, more than one species for some genera have been sequenced in liverworts (Calypogeia), mosses (Bartramia, Climacium, Niphotrichum, Orthotrichum, Racomitrium, and Stoneobryum), hornworts (Anthoceros), and monocot angiosperms (Oryza) and eudicot angiosperms (Beta, Brassica, Corchorus, Glycine, Gossypium, Helianthus, Nicotiana, Populus, Salix, Senna, Solanum, Sorbus, Spondias, and Trifolium). The authors could have done an inter-species mitogenome rearrangement, intron, repeat sequence comparison among all these genera, and then plotted the result along a phylogenetic gradient of land plants. Any emerging pattern will be more meaningful than the sort of comparison they have reported on genome rearrangement, gene and intron content, and repeat in the manuscript.

A minor point that the authors could have done more analysis is about RNA editing (Lines 413-415 in their manuscript). They should check to see how their results conform to the editing pattern shaped by selection as suggested by Jobson and Qiu (2008).

6. PLOS authors have the option to publish the peer review history of their article (what does this mean?). If published, this will include your full peer review and any attached files.

Reviewer #1: **Yes: **Yin-Long Qiu

---

## [Author Response · Author response to Decision Letter 0]

3 Jul 2021

Reviewer #1: Review of “The complete mitochondrial genome of Cycas debaoensis revealed unexpected static evolution in gymnosperm species”

It is nice to have the second Cycas species sequenced for the mitochondrial genome, as it provides data to gain insights into mitogenome evolution in closely related species in this early diverging lineage of gymnosperms. The authors have done a good job of describing the genome structure and content, and have also done a somewhat informative comparison on genome content between cycads and other land plants.

One major point that the authors have failed to appreciate is that the second Cycas mitogenome provides an excellent opportunity to demonstrate the amplitude of genome rearrangement among closely related species in different lineages of land plants, where the genome embarked on a path of radical structural evolution, among all eukaryotes. Until now, more than one species for some genera have been sequenced in liverworts (Calypogeia), mosses (Bartramia, Climacium, Niphotrichum, Orthotrichum, Racomitrium, and Stoneobryum), hornworts (Anthoceros), and monocot angiosperms (Oryza) and eudicot angiosperms (Beta, Brassica, Corchorus, Glycine, Gossypium, Helianthus, Nicotiana, Populus, Salix, Senna, Solanum, Sorbus, Spondias, and Trifolium). The authors could have done an inter-species mitogenome rearrangement, intron, repeat sequence comparison among all these genera, and then plotted the result along a phylogenetic gradient of land plants. Any emerging pattern will be more meaningful than the sort of comparison they have reported on genome rearrangement, gene and intron content, and repeat in the manuscript. 

Response: We have done an inter-species mitogenome rearrangement, intron, repeat sequence comparison as suggested by the reviewer. Inter-species mitogenome rearrangement analysis includes 46 taxa with enough representation from all the major lineages of land plants. Number of repeats from each observed taxa are also mentioned along a phylogenetic gradient of land plants (Fig3.eps). Furthermore, distribution pattern of 26 ancestral seed plant introns among major land plant groups has also been investigated and discussed (Fig2.eps).

A minor point that the authors could have done more analysis is about RNA editing (Lines 413-415 in their manuscript). They should check to see how their results conform to the editing pattern shaped by selection as suggested by Jobson and Qiu (2008).

Response: we explored whether the editing frequencies in gymnosperm species are shaped by selection constraint on genes as suggested by Jobson and Qiu (2008). Three gymnosperm taxa included in the analysis were C. debaoensis, G. biloba and T. cuspidata. We found that membrane-bounded and soluble protein coding genes have experienced similar selective pressures as there is no clear pattern among editing efficiency and gene substitution rates (dN and dS vs. editing efficiency %) of three observed gymnosperm taxa (S7 Table).

All the relevant details are mentioned in revised manuscript.

---

## [Decision Letter · Decision Letter 1]

12 Jul 2021

The complete mitochondrial genome of Cycas debaoensis revealed unexpected static evolution in gymnosperm species

PONE-D-21-15488R1

Dear Dr. Zhang,

We’re pleased to inform you that your manuscript has been judged scientifically suitable for publication and will be formally accepted for publication once it meets all outstanding technical requirements.

Kind regards,

Tzen-Yuh Chiang

Academic Editor

PLOS ONE

Additional Editor Comments (optional):

Reviewers' comments:

Reviewer's Responses to Questions

**Comments to the Author**

1. If the authors have adequately addressed your comments raised in a previous round of review and you feel that this manuscript is now acceptable for publication, you may indicate that here to bypass the “Comments to the Author” section, enter your conflict of interest statement in the “Confidential to Editor” section, and submit your "Accept" recommendation.

Reviewer #1: All comments have been addressed

2. Is the manuscript technically sound, and do the data support the conclusions?

Reviewer #1: Yes

3. Has the statistical analysis been performed appropriately and rigorously? 

Reviewer #1: Yes

4. Have the authors made all data underlying the findings in their manuscript fully available?

Reviewer #1: Yes

5. Is the manuscript presented in an intelligible fashion and written in standard English?

Reviewer #1: Yes

6. Review Comments to the Author

Reviewer #1: (No Response)

7. PLOS authors have the option to publish the peer review history of their article (what does this mean?). If published, this will include your full peer review and any attached files.

Reviewer #1: No

---

## [Editor Report · Acceptance letter]

14 Jul 2021

PONE-D-21-15488R1 

The complete mitochondrial genome of *Cycas debaoensis* revealed unexpected static evolution in gymnosperm species 

Dear Dr. Zhang:

I'm pleased to inform you that your manuscript has been deemed suitable for publication in PLOS ONE. Congratulations! Your manuscript is now with our production department. 

Kind regards, 

on behalf of

Dr. Tzen-Yuh Chiang 

Academic Editor

PLOS ONE